# Unified Perspectives on Layer Balancing and Parameter-norm Evolution in Neural Nets

**Jasraj Singh**[*]  **Enea Monzio Compagnoni**  **Antonio Orvieto**
Universität Basel                    ELLIS Institute, Tübingen
                    Max Planck Institute for Intelligent Systems

## Abstract

Understanding the parameter dynamics under gradient-based training has been central to explaining implicit regularization and generalization in deep learning, with balancedness of layers – defined as the difference between the left Gramian of a layer and the right Gramian of the next layer – playing a key role in many existing analyses. We present a unified and substantially more general framework for studying layer-balancedness and parameter-norm dynamics across a broad class of neural architectures. Modeling networks as compositions of learnable Hilbert-Schmidt operators interleaved with fixed positive-homogeneous nonlinearities, we show that consecutive layers without nonlinearities in-between converge exponentially fast toward a balanced state under weight decay. Furthermore, we derive a general expression for the time evolution of the squared-norm of each learnable layer, showing that parameter-norm dynamics reduce to a single scalar quantity: the inner product between the network output and the negative gradient of the loss with respect to it. Our framework recovers existing results as special cases while extending them to architectures beyond the reach of prior, architecture-specific analyses. Finally, it connects parameter evolution to function-space dynamics, which can be studied, for example, using the NTK theory and mean-field analysis.

## 1 Introduction

The dynamics of a neural network's parameters under gradient-based training have received sustained attention as a lens into implicit regularization and generalization in deep learning. Prior works have established layer-wise balancing (Arora et al., 2018; Cao et al., 2025; Du et al., 2018), which has played an important role in studying the training dynamics of linear networks (Arora et al., 2018; 2019a), global optimality in shallow matrix factorization problems (Du et al., 2018), low-rank bias in shallow (Min et al., 2021; Tu et al., 2024) and deep matrix factorization (Arora et al., 2019b), richness of learning in deep linear networks (Dominé et al., 2024), and neural collapse (Jacot et al., 2025). These analyses of layer-wise balancing, however, are typically tied to architecture-specific arguments that do not extend to practical components of modern neural networks.

In this work, we provide a unified and substantially more general framework for studying the dynamics of the balance between consecutive layers (Equation 3), as well as layer-wise parameter-norms. We model neural networks as compositions of learnable Hilbert-Schmidt operators – which includes fully connected layers and convolutions – interleaved with fixed positive-homogeneous nonlinear operators, including common coordinate-wise activation functions, pooling layers, and normalization layers. Studying such models under gradient flow, in Theorem 1, we show that consecutive layers with no nonlinearity between them become balanced at an exponential rate proportional to the weight decay factor. In Theorem 2, we derive a general expression for the time evolution of the squared norm of every learnable layer, showing that it reduces to a single scalar quantity: $\langle \mathbf{f}_{\boldsymbol{\theta}}(\mathbf{X}), -\nabla_{\mathbf{f}} \mathcal{L}(\mathbf{f}_{\boldsymbol{\theta}}(\mathbf{X}); \mathbf{Y}) \rangle$. Overall, our framework immediately recovers results in the literature, while also including nontrivial architectures beyond the scope of existing analyses.

---

[*]Work done during an internship at the ELLIS Institute, Tübingen, and Max Planck Institute for Intelligent Systems. Correspondence to <jasraj.singh00150@gmail.com>.

## 2 SETUP

Consider a sequence of Hilbert-Schmidt operators, $\boldsymbol{\theta} := \left\{ \mathcal{T}_\ell \in \mathrm{HS}\left(\mathcal{H}_{\ell-1}, \mathcal{H}_\ell\right) \right\}_{\ell=1}^L$, where $\mathcal{H}$ denotes a Hilbert space. We define an $L$-layer neural network recursively as follows:

$$\text{pre-activations:} \quad \mathbf{Z}_\ell := \mathcal{T}_\ell\left(\mathbf{A}_{\ell-1}\right) \tag{1a}$$

$$\text{post-activations:} \quad \mathbf{A}_\ell := \phi_\ell\left(\mathbf{Z}_\ell\right) \tag{1b}$$

for hidden layers $\ell \in [L-1]$, with inputs given by $\mathbf{A}_0 := \mathbf{X} \in \mathcal{H}_0$ and output by $\mathbf{f}_{\boldsymbol{\theta}}\left(\mathbf{X}\right) := \mathbf{Z}_L = \mathcal{T}_L\left(\mathbf{A}_{L-1}\right)$; $\mathcal{T}_\ell$ are learnable linear operators while $\phi_\ell : \mathcal{H}_\ell \to \mathcal{H}_\ell$ are fixed nonlinear operators.

**Assumption 1.** *The nonlinearities, $\phi_\ell$, are Fréchet differentiable at their inputs, $\mathbf{Z}_\ell$.*

We consider model training with gradient flow (GF):

$$\dot{\boldsymbol{\theta}} = -\nabla_{\boldsymbol{\theta}} \mathcal{L}\left(\mathbf{f}_{\boldsymbol{\theta}}\left(\mathbf{X}\right); \mathbf{Y}\right) - \delta \cdot \nabla_{\boldsymbol{\theta}} \mathcal{R}\left(\boldsymbol{\theta}\right) \tag{2}$$

where $\mathbf{X} \in \mathcal{H}_0$ collects the inputs, $\mathbf{Y} \in \mathcal{H}_L$ collects the targets, $\boldsymbol{\theta}$ parameterizes the model, $\mathcal{L}$ is a scalar-valued loss function that is minimized, $\mathcal{R}$ is a regularizer, and $\delta$ is the regularization factor.

**Note:** Going forward, we assume that the loss is the same at each step, for the sake of clarity. This is an unnecessary assumption for *all our results*, which can be trivially extended to the case of time-dependent loss functions, e.g. stochastic mini-batch optimization. Similarly, we use Tikhonov regularization for simplicity, but other regularizers may yield analogous results.

## 3 DYNAMICS OF BALANCEDNESS

Several different definitions of balancedness between two consecutive layers have been proposed in the literature (Arora et al., 2019b; Dominé et al., 2024; Du et al., 2018; Tu et al., 2024). We keep our analysis general by studying the mathematical object common to all of these definitions:

$$\mathcal{B}\left(\mathcal{T}_\ell, \mathcal{T}_{\ell+1}\right) := \mathcal{T}_\ell \circ \mathcal{T}_\ell^\star - \mathcal{T}_{\ell+1}^\star \circ \mathcal{T}_{\ell+1} \tag{3}$$

where $\circ$ denotes composition of operators, and $\cdot^\star$ denotes the adjoint of an operator. Intuitively, near-zero $\mathcal{B}\left(\mathcal{T}_\ell, \mathcal{T}_{\ell+1}\right)$ suggests that the layers are approximately balanced.

**Theorem 1.** *Consider an $L$-layer neural network with the nonlinearities $\phi_\ell$ satisfying Assumption 1. With $\phi_\ell$ being the identity map, under gradient flow training,*

$$\mathcal{B}\left(\mathcal{T}_\ell\left(t\right), \mathcal{T}_{\ell+1}\left(t\right)\right) = \exp\left(-2\delta t\right) \cdot \mathcal{B}\left(\mathcal{T}_\ell\left(0\right), \mathcal{T}_{\ell+1}\left(0\right)\right) \tag{4}$$

This result was proposed in Arora et al. (2018, Theorem 1) for linear Multilayered Perceptrons (MLPs) initialized as $\mathcal{B}\left(\mathcal{T}_\ell\left(t\right), \mathcal{T}_{\ell+1}\left(t\right)\right) = 0$, i.e. they prove that networks remain balanced if initialized as such. Du et al. (2018, Theorem 2.2) proposed a more general result for MLPs trained *without weight decay*, requiring only $\phi_\ell$ to be the identity map instead of the whole network, same as Theorem 1.

**Definition 1.** *A positive-homogeneous function of order $k$ satisfies $\phi\left(c\mathbf{x}\right) = c^k \phi\left(\mathbf{x}\right)$, where $c \in \mathbb{R}_{++}$. For such functions, Euler's homogeneous function theorem states that $\left[\mathrm{D}\phi\left(\mathbf{x}\right)\right]\left(\mathbf{x}\right) = k\phi\left(\mathbf{x}\right)$ whenever $\phi$ is differentiable at $\mathbf{x}$.*

**Theorem 2.** *Consider an $L$-layer neural network with the nonlinearities $\phi_\ell$ satisfying Assumption 1. With positive-homogeneous nonlinearities $\phi_\ell$ of order $k_\ell$, under gradient flow training,*

$$\frac{d}{dt}\|\mathcal{T}_\ell\|_{\mathrm{HS}(\mathcal{H}_{\ell-1}, \mathcal{H}_\ell)}^2 = -2\left(\prod_{l=\ell}^L k_l\right) \cdot \left\langle \nabla_{\mathbf{f}} \mathcal{L}\left(\mathbf{f}_{\boldsymbol{\theta}}\left(\mathbf{X}\right); \mathbf{Y}\right), \mathbf{f}_{\boldsymbol{\theta}}\left(\mathbf{X}\right)\right\rangle_{\mathcal{H}_L} - 2\delta \cdot \|\mathcal{T}_\ell\|_{\mathrm{HS}(\mathcal{H}_{\ell-1}, \mathcal{H}_\ell)}^2 \tag{5}$$

*with $k_L := 1$.*

Cao et al. (2025, Theorem 1) proposed this result for the case of scale-invariant tensorized models, which corresponds to the $k_\ell = 1$ setting. Moreover, for the special cases of MLPs and Convolutional Neural Networks (CNNs), Du et al. (2018, Corollary 2.1 and Theorem 2.3) proposed that the difference in norms of consecutive layers with order-1 coordinate-wise activations between them, doesn't change; note that they don't consider weight decay, i.e. $\delta = 0$. This can be derived by setting $k_\ell = 1$ – or, more trivially, having one of $\phi_{\ell+1}, \phi_{\ell+2}, \ldots, \phi_{L-1}$ being order-0 homogeneous – in Theorem 2.

**Note:** In Theorem 2, the nonlinear operators are simply positive-homogeneous. Hence, it includes non-trivial deep learning modules, like pooling layers and normalization layers.

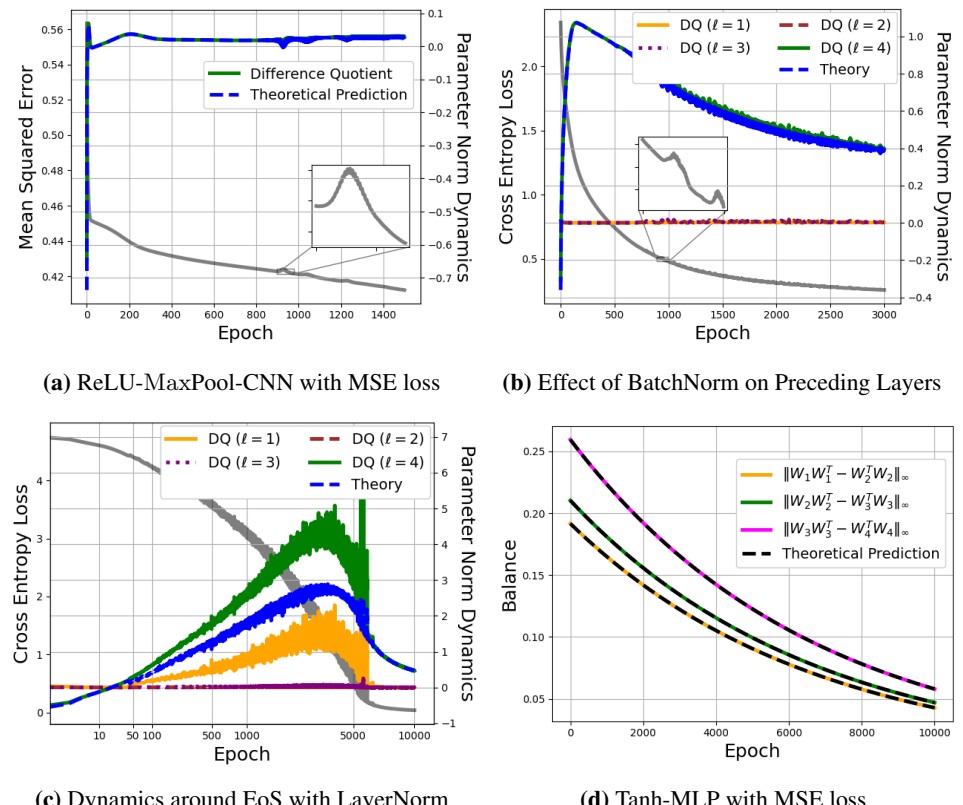

**(a)** ReLU-MaxPool-CNN with MSE loss

**(b)** Effect of BatchNorm on Preceding Layers

**(c)** Dynamics around EoS with LayerNorm

**(d)** Tanh-MLP with MSE loss

**Figure 1:** Theoretical predictions (Theorem 1 and Theorem 2) and empirical dynamics (Equation 32).

**Corollary 1.** *Under the setup in Theorem 2,*

$$\frac{d}{dt} \mathrm{Tr}\left(\mathcal{B}\left(\mathcal{T}_\ell\left(t\right), \mathcal{T}_{\ell+1}\left(t\right)\right)\right) = -2\left(k_\ell - 1\right)\left(\prod_{l=\ell+1}^{L} k_l\right) \cdot \left\langle \nabla_{\mathbf{f}} \mathcal{L}\left(\mathbf{f}_{\boldsymbol{\theta}}\left(\mathbf{X}\right); \mathbf{Y}\right), \mathbf{f}_{\boldsymbol{\theta}}\left(\mathbf{X}\right) \right\rangle_{\mathcal{H}_L} \tag{6}$$
$$- 2\delta \cdot \mathrm{Tr}\left(\mathcal{B}\left(\mathcal{T}_\ell\left(t\right), \mathcal{T}_{\ell+1}\left(t\right)\right)\right)$$

## 4 SPECIAL CASES IN EUCLIDEAN SPACES

Consider $\{\mathcal{H}_\ell\}_{\ell=0}^L$ to be a sequence of real vector spaces with finite dimensions, $\{d_\ell\}_{\ell=0}^L$. Upon fixing bases of $\mathcal{H}_{\ell-1}$ and $\mathcal{H}_\ell$, the linear map $\mathcal{T}_\ell : \mathcal{H}_{\ell-1} \to \mathcal{H}_\ell$ admits a unique matrix representation $\mathbf{W}_\ell \in \mathbb{R}^{d_\ell \times d_{\ell-1}}$, which we treat as a learnable weight matrix. In this case, we have

$$\|\mathcal{T}_\ell\|_{\mathrm{HS}(\mathcal{H}_{\ell-1}, \mathcal{H}_\ell)} = \|\mathbf{W}_\ell\|_F \tag{7}$$

where $\|\cdot\|_F$ is the Frobenius norm. Hence, the model-parameters' Frobenius norm's dynamics are given by Theorem 2. In particular, this includes full-connected layers and convolution layers.

**Lemma 1.** *A positive-homogeneous function, $\phi_k\left(\cdot; c_-, c_0, c_+\right) : \mathbb{R} \to \mathbb{R}$, of order $k$ takes the form*

$$\phi_k\left(\cdot; c_-, c_0, c_+\right) = \begin{cases} c_-\left(-x\right)^k, & x < 0 \\ c_0, & x = 0 \\ c_+ x^k, & x > 0 \end{cases} \tag{8}$$

*where $c_\pm = \phi\left(\pm 1\right) \in \mathbb{R}$, and $c_0 = \phi\left(0\right) \in \mathbb{R}$. Moreover, if $k \neq 0$, then $c_0 = 0$.*

From Lemma 1, we can immediately identify the following order-1 positive-homogeneous activations: the identity map, ReLU activation, and LeakyReLU activation.

**Lemma 2.** *A sequence of positively homogeneous operators can be replaced with one whose homogeneity order is the product of the homogeneity orders of the component operators.*

**Pooling Layers.** Max pooling and Avg pooling layers in a CNN – as well as analogous operators, like pooling layers in graph neural networks (Grattarola et al., 2024) – are order-1 homogeneous. In light of Lemma 2, the result in Theorem 2 remains unaffected if pooling layers are used.

**Normalization Layers.** Consider the common normalization techniques used in neural network training, e.g. BatchNorm (Ioffe & Szegedy, 2015), LayerNorm (Ba et al., 2016), InstanceNorm (Ulyanov et al., 2017), GroupNorm (Wu & He, 2018), RMSNorm (Zhang & Sennrich, 2019), $\ell^p$ norm strategies (Hoffer et al., 2018; Santurkar et al., 2018), etc. These operators subtract the mean along certain dimensions of the layer input, and then divide by the standard deviation.[1] When implemented using the input's statistics and not running averages, these operators are order-0 homogeneous, i.e. their output is invariant to input scaling. Therefore, normalization layers (and order-0 positive-homogeneous operators, in general) result in the sizes of all previous layers to decay at an exponential rate proportional to $\delta$, effectively having an L2-regularization-like effect.

A similar conclusion was made in Liu et al. (2022, Corollary 3.2) for the specific case of ReLU-MLPs with BatchNorm. While their result was limited to the trainable layer immediately preceding BatchNorm, ours extends this to *all preceding layers*.

## 5    Empirical Evidence

Figure 1a show the parameter-norm dynamics of a ReLU-CNN with Max pooling after every layer, trained with Mean Squared-Error (MSE) loss – we can see a strong agreement between the theoretical predictions and the empirical approximations made using difference quotients. Figure 1b shows the parameter-norm dynamics of a LeakyReLU-CNN, with BatchNorm applied only before the readout, trained with cross-entropy (CE) loss. We see that the norms of *all* preceding layers stay constant, and the network's norm-dynamics are governed by those of the last layer.

### 5.1    Breakdown at the edge-of-stability

Of course, our results for GF don't translate over to gradient descent (GD) without limitations, since the two optimizers' trajectories don't always coincide. Specifically, our predictions are precise only until the *edge-of-stability* (EoS), i.e. until the sharpness – defined as the maximum eigenvalue of the loss Hessian – is below $2/\eta$ (Cohen et al., 2021). In this regime, the GD trajectory *often* coincides with the GF trajectory, as noted in Cohen et al. (2021, Section 3.4). However, once sharpness crosses this threshold, it stops tracking GF, and our theory consistently *underestimates* the rate of growth of parameter-norm, as can be seen in mid-training in Figure 1c.

### 5.2    Robustness to Choice of Activation

Surprisingly, theoretical predictions with $k_\ell = 1$ are quite precise for Tanh networks trained with MSE loss, even though the activation function is *not* positively homogeneous, so neither of our theorems apply. In Figure 1d, we show that balance between layers – more specifically, the maximum absolute value of $\mathbf{W}_\ell \mathbf{W}_\ell^\top - \mathbf{W}_{\ell+1}^\top \mathbf{W}_{\ell+1}$ – decays at the rate predicted in Theorem 1. While the same does not apply to CE loss, except in early-training, as can be seen in Figure 2b, the norm-dynamics are qualitatively similar to the theoretical predictions, as seen in Figure 2a. A partial explanation for this observation is that Tanh can be approximated by the identity map ($k = 1$) for small pre-activations.

## 6    Conclusion

Our work provides a unified framework for analyzing parameter-norm dynamics and layer balancedness across a broad range of neural architectures. We show that the evolution of layer norms can be reduced to a single scalar quantity linked to the network's output and loss gradient. Our results on exponential convergence toward balancedness under weight decay explain how layers synchronize during training, even when starting from unbalanced initializations. Our results may allow parameter evolution to be studied deeper through the lens of the network's function evolution, on which there has been extensive research, especially in the infinite-width limit via neural tangent kernels (Jacot et al., 2018; Lee et al., 2019) and mean-field dynamics (Chizat & Bach, 2018).

---

[1]Normalization is optionally followed by an affine transform, which we assume does not include a shift term. In this case, the layer can be thought of as a composition of a linear map and an order-0 nonlinearity.

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

## A  NOTATION AND MATHEMATICAL BACKGROUND

With perhaps a few exceptions, we use the following conventions:

- $\dot{\boldsymbol{\theta}}$ denotes the time-derivative of the parameters, $\boldsymbol{\theta}$.

- $\mathcal{H}$ denote Hilbert spaces, and $\mathcal{T}$ denote Hilbert-Schmidt operators.

- Bold-faced capital letters denote members of some Hilbert space, e.g. $\mathbf{A}$, $\mathbf{G}$, $\mathbf{X}$, $\mathbf{Y}$, and $\mathbf{Z}$.

- Normal-font letters denote real-valued scalars, e.g. $c$, $H$, $k$, $\ell$, $L$, $n$, $t$, $x$, $\delta$, and $\eta$.

**Definition 2** (Hilbert Space). *A Hilbert space is a vector space equipped with an inner product, complete with respect to the induced norm.*

**Definition 3** (Hilbert-Schmidt Norm). *The Hilbert-Schmidt norm of a linear operator $\mathcal{T} : \mathcal{H}_1 \to \mathcal{H}_2$ is defined as*

$$\|\mathcal{T}\|_{\mathrm{HS}(\mathcal{H}_1,\mathcal{H}_2)}^2 \coloneqq \sum_{i\in\mathcal{I}} \|\mathcal{T}\mathbf{e}_i\|_{\mathcal{H}_2}^2 \tag{9}$$

*where $\{\mathbf{e}_i : i \in \mathcal{I}\}$ is any orthonormal basis of $\mathcal{H}_1$ and $\mathcal{I}$ is a corresponding (countable) index set.*

The Hilbert-Schmidt norm is well-defined, i.e. it is independent of the choice of orthonormal basis. If $\mathcal{H}_1 \coloneqq \mathbb{R}^m$ and $\mathcal{H}_2 \coloneqq \mathbb{R}^n$, and $\mathcal{T} = \mathbf{W} \in \mathbb{R}^{m\times n}$, then it is easy to see that the Hilbert-Schmidt norm coincides with the Frobenius norm.

**Definition 4** (Hilbert-Schmidt Operator). *A Hilbert-Schmidt operator is a linear operator, $\mathcal{T} : \mathcal{H}_1 \to \mathcal{H}_2$, with finite Hilbert-Schmidt norm.*

**Definition 5** (Inner-Product and Trace). *For two Hilbert-Schmidt operators, $\mathcal{T}_1 : \mathcal{H}_1 \to \mathcal{H}_2$ and $\mathcal{T}_2 : \mathcal{H}_1 \to \mathcal{H}_2$, the Hilbert-Schmidt inner-product is defined as*

$$\langle \mathcal{T}_1, \mathcal{T}_2 \rangle_{\mathrm{HS}(\mathcal{H}_1,\mathcal{H}_2)} \coloneqq \sum_{i\in\mathcal{I}} \langle \mathcal{T}_1\mathbf{e}_i, \mathcal{T}_2\mathbf{e}_i \rangle_{\mathcal{H}_2} \tag{10}$$

*The trace operator is defined as $\mathrm{Tr}\left(\mathcal{T}_2^\star \circ \mathcal{T}_1\right) = \langle \mathcal{T}_1, \mathcal{T}_2 \rangle_{\mathrm{HS}(\mathcal{H}_1,\mathcal{H}_2)}$.*

**Definition 6** (Outer-Product). *For $\mathbf{u} \in \mathcal{H}_1$ and $\mathbf{v} \in \mathcal{H}_2$, we define the outer-product, $\mathbf{u} \otimes \mathbf{v} : \mathcal{H}_2 \to \mathcal{H}_1$, as the map $\mathbf{w} \mapsto \langle \mathbf{v}, \mathbf{w} \rangle_{\mathcal{H}_2} \mathbf{u}$, with $\mathbf{w} \in \mathcal{H}_2$. In particular, $\mathbf{u} \otimes \mathbf{v}$ is a Hilbert-Schmidt operator.*

**Definition 7** (Adjoint Operator). *The Hermition adjoint (or conjugate) of a linear operator, $\mathcal{T} : \mathcal{H}_1 \to \mathcal{H}_2$, is the unique linear operator, $\mathcal{T}^\star : \mathcal{H}_2 \to \mathcal{H}_1$, that satisfies*

$$\langle \mathcal{T}\mathbf{x}_1, \mathbf{x}_2 \rangle_{\mathcal{H}_2} = \langle \mathbf{x}_1, \mathcal{T}^\star\mathbf{x}_2 \rangle_{\mathcal{H}_1}, \quad \forall \mathbf{x}_1 \in \mathcal{H}_1, \ \forall \mathbf{x}_2 \in \mathcal{H}_2 \tag{11}$$

**Definition 8** (Fréchet Derivative). *Say $\mathcal{U}$ is an open subset of $\mathcal{H}_1$. A map $\mathcal{F} : \mathcal{U} \to \mathcal{H}_2$ is Fréchet differentiable at $\mathbf{x} \in \mathcal{U}$ if there exists a bounded linear operator, $\mathrm{D}\mathcal{F}(\mathbf{x}) : \mathcal{U} \to \mathcal{H}_2$, such that*

$$\lim_{\|\mathbf{h}\|_{\mathcal{H}_1}\to 0} \frac{\|\mathcal{F}(\mathbf{x}+\mathbf{h}) - \mathcal{F}(\mathbf{x}) - \mathrm{D}\mathcal{F}(\mathbf{x})\mathbf{h}\|_{\mathcal{H}_2}}{\|\mathbf{h}\|_{\mathcal{H}_1}} = 0 \tag{12}$$

**Theorem 3** (Riesz Representation). *For every continuous linear functional $\mathcal{T} : \mathcal{H} \to \mathbb{R}$, there exists a unique element $\mathbf{x} \in \mathcal{H}$ such that $\forall \mathbf{y} \in \mathcal{H}$, we have $\mathcal{T}(\mathbf{y}) = \langle \mathbf{x}, \mathbf{y} \rangle_{\mathcal{H}}$.*

In particular, if $\mathcal{F} : \mathcal{H} \to \mathbb{R}$ is a scalar-valued function, then the Riesz representation of its derivative at some point, $\mathrm{D}\mathcal{F}(\mathbf{x}) : \mathcal{H} \to \mathbb{R}$, is called the *gradient*, denoted by $\nabla\mathcal{F}(\mathbf{x})$.

Say $\mathbf{x} = (\mathbf{x}_1, \mathbf{x}_2) \in \mathcal{H}_1 \times \mathcal{H}_2 = \mathcal{H}$. We denote by $\mathrm{D}_{\mathbf{x}_1}\mathcal{F}(\mathbf{x}) : \mathcal{H}_1 \to \mathbb{R}$ the *partial derivative* with respect to $\mathbf{x}_1$ (keeping $\mathbf{x}_2$ fixed), and by $\nabla_{\mathbf{x}_1}\mathcal{F}(\mathbf{x})$ its Riesz representation.

## B  PROOFS

**Lemma 3.** *Consider an L-layer neural network with the nonlinearities $\phi_\ell$ satisfying Assumption 1. Under gradient flow training, $\forall \ell \in [L-1]$, the evolution of the left-Gramian is given by*

$$\frac{d}{dt}\left(\mathcal{T}_\ell \circ \mathcal{T}_\ell^\star\right) = \left[-\mathbf{G}_\ell \otimes \mathbf{Z}_\ell - \delta \cdot \mathcal{T}_\ell \circ \mathcal{T}_\ell^\star\right] + h.c. \tag{13}$$

*while that of the right-Gramian is given by*

$$\frac{d}{dt}\left(\mathcal{T}_{\ell+1}^{\star}\circ\mathcal{T}_{\ell+1}\right) = \left[-\mathcal{T}_{\ell+1}^{\star}\left(\mathbf{G}_{\ell+1}\right)\otimes\mathbf{A}_{\ell} - \delta\cdot\mathcal{T}_{\ell+1}^{\star}\circ\mathcal{T}_{\ell+1}\right] + \text{h.c.} \tag{14}$$

*where* $\mathbf{G}_{\ell} \coloneqq \nabla_{\mathbf{Z}_{\ell}}\mathcal{L} \in \mathcal{H}_{\ell}$, *and h.c. denotes the Hermitian-conjugate of the preceding term.*

**Proof.** Under [Assumption 1](), the evolution of left-Gramian of $\mathcal{T}_{\ell}$ is given by

$$\frac{\mathrm{d}}{\mathrm{d}t}\left(\mathcal{T}_{\ell}\circ\mathcal{T}_{\ell}^{\star}\right) = \left(\frac{\mathrm{d}}{\mathrm{d}t}\mathcal{T}_{\ell}\right)\circ\mathcal{T}_{\ell}^{\star} + \text{h.c.} \tag{15a}$$

$$= \left(-\mathbf{G}_{\ell}\otimes\mathbf{A}_{\ell-1} - \delta\cdot\mathcal{T}_{\ell}\right)\circ\mathcal{T}_{\ell}^{\star} + \text{h.c.} \tag{15b}$$

$$= \left[-\mathbf{G}_{\ell}\otimes\mathcal{T}_{\ell}\left(\mathbf{A}_{\ell-1}\right) - \delta\cdot\mathcal{T}_{\ell}\circ\mathcal{T}_{\ell}^{\star}\right] + \text{h.c.} \tag{15c}$$

$$= \left[-\mathbf{G}_{\ell}\otimes\mathbf{Z}_{\ell} - \delta\cdot\mathcal{T}_{\ell}\circ\mathcal{T}_{\ell}^{\star}\right] + \text{h.c.} \tag{15d}$$

while that of the right-Gramian of $\mathcal{T}_{\ell+1}$ is given by

$$\frac{\mathrm{d}}{\mathrm{d}t}\left(\mathcal{T}_{\ell+1}^{\star}\circ\mathcal{T}_{\ell+1}\right) = \mathcal{T}_{\ell+1}^{\star}\circ\left(\frac{\mathrm{d}}{\mathrm{d}t}\mathcal{T}_{\ell+1}\right) + \text{h.c.} \tag{16a}$$

$$= \mathcal{T}_{\ell+1}^{\star}\circ\left(-\mathbf{G}_{\ell+1}\otimes\mathbf{A}_{\ell} - \delta\cdot\mathcal{T}_{\ell+1}\right) + \text{h.c.} \tag{16b}$$

$$= \left[-\mathcal{T}_{\ell+1}^{\star}\left(\mathbf{G}_{\ell+1}\right)\otimes\mathbf{A}_{\ell} - \delta\cdot\mathcal{T}_{\ell+1}^{\star}\circ\mathcal{T}_{\ell+1}\right] + \text{h.c.} \tag{16c}$$

**Theorem 1.** *Consider an L-layer neural network with the nonlinearities $\phi_{\ell}$ satisfying [Assumption 1](). With $\phi_{\ell}$ being the identity map, under gradient flow training,*

$$\mathcal{B}\left(\mathcal{T}_{\ell}\left(t\right),\mathcal{T}_{\ell+1}\left(t\right)\right) = \exp\left(-2\delta t\right)\cdot\mathcal{B}\left(\mathcal{T}_{\ell}\left(0\right),\mathcal{T}_{\ell+1}\left(0\right)\right) \tag{4}$$

**Proof.** The *pre-gradient* satisfies the recursion

$$\mathbf{G}_{\ell} = \left[\mathrm{D}\phi_{\ell}\left(\mathbf{Z}_{\ell}\right)\right]^{\star}\left(\mathcal{T}_{\ell+1}^{\star}\left(\mathbf{G}_{\ell+1}\right)\right) \tag{17}$$

If $\phi_{\ell}$ is the identity map, then $\mathbf{G}_{\ell} = \mathcal{T}_{\ell+1}^{\star}\left(\mathbf{G}_{\ell+1}\right)$ and $\mathbf{A}_{\ell} = \mathbf{Z}_{\ell}$, so using [Lemma 3](),

$$\frac{\mathrm{d}}{\mathrm{d}t}\mathcal{B}\left(\mathcal{T}_{\ell}\left(t\right),\mathcal{T}_{\ell+1}\left(t\right)\right) = -2\delta\cdot\mathcal{B}\left(\mathcal{T}_{\ell}\left(t\right),\mathcal{T}_{\ell+1}\left(t\right)\right) \tag{18a}$$

$$\implies \mathcal{B}\left(\mathcal{T}_{\ell}\left(t\right),\mathcal{T}_{\ell+1}\left(t\right)\right) = \exp\left(-2\delta t\right)\cdot\mathcal{B}\left(\mathcal{T}_{\ell}\left(0\right),\mathcal{T}_{\ell+1}\left(0\right)\right) \tag{18b}$$

**Theorem 2.** *Consider an L-layer neural network with the nonlinearities $\phi_{\ell}$ satisfying [Assumption 1](). With positive-homogeneous nonlinearities $\phi_{\ell}$ of order $k_{\ell}$, under gradient flow training,*

$$\frac{d}{dt}\|\mathcal{T}_{\ell}\|_{\mathrm{HS}(\mathcal{H}_{\ell-1},\mathcal{H}_{\ell})}^{2} = -2\left(\prod_{l=\ell}^{L}k_{l}\right)\cdot\left\langle\nabla_{\mathbf{f}}\mathcal{L}\left(\mathbf{f}_{\boldsymbol{\theta}}\left(\mathbf{X}\right);\mathbf{Y}\right),\mathbf{f}_{\boldsymbol{\theta}}\left(\mathbf{X}\right)\right\rangle_{\mathcal{H}_{L}} - 2\delta\cdot\|\mathcal{T}_{\ell}\|_{\mathrm{HS}(\mathcal{H}_{\ell-1},\mathcal{H}_{\ell})}^{2} \tag{5}$$

*with* $k_{L} \coloneqq 1$.

**Proof.** Under [Assumption 1](), the parameter-norm dynamics are given by

$$\frac{\mathrm{d}}{\mathrm{d}t}\|\mathcal{T}_{\ell}\|_{\mathrm{HS}(\mathcal{H}_{\ell-1},\mathcal{H}_{\ell})}^{2} = 2\cdot\left\langle\mathcal{T}_{\ell},-\nabla_{\mathcal{T}_{\ell}}\mathcal{L}\right\rangle_{\mathrm{HS}(\mathcal{H}_{\ell-1},\mathcal{H}_{\ell})} + 2\cdot\left\langle\mathcal{T}_{\ell},-\delta\cdot\mathcal{T}_{\ell}\right\rangle_{\mathrm{HS}(\mathcal{H}_{\ell-1},\mathcal{H}_{\ell})} \tag{19a}$$

$$= 2\cdot\left\langle\mathcal{T}_{\ell},-\nabla_{\mathcal{T}_{\ell}}\mathcal{L}\right\rangle_{\mathrm{HS}(\mathcal{H}_{\ell-1},\mathcal{H}_{\ell})} - 2\delta\cdot\|\mathcal{T}_{\ell}\|_{\mathrm{HS}(\mathcal{H}_{\ell-1},\mathcal{H}_{\ell})}^{2} \tag{19b}$$

We will drop the contribution from the regularizer for clarity; it can be picked back up in the final result. $\forall\ell\in[L]$, using [Lemma 3](), we have

$$\frac{\mathrm{d}}{\mathrm{d}t}\|\mathcal{T}_{\ell}\|_{\mathrm{HS}(\mathcal{H}_{\ell-1},\mathcal{H}_{\ell})}^{2} = \frac{\mathrm{d}}{\mathrm{d}t}\mathrm{Tr}\left(\mathcal{T}_{\ell}\circ\mathcal{T}_{\ell}^{\star}\right) \tag{20a}$$

$$= -2\cdot\left\langle\mathbf{G}_{\ell},\mathbf{Z}_{\ell}\right\rangle_{\mathcal{H}_{\ell}} \tag{20b}$$

Moreover, using the recursion in [Equation 17](),

$$\left\langle\mathbf{G}_{\ell},\mathbf{Z}_{\ell}\right\rangle_{\mathcal{H}_{\ell}} = \left\langle\left[\mathrm{D}\phi_{\ell}\left(\mathbf{Z}_{\ell}\right)\right]^{\star}\left(\mathcal{T}_{\ell+1}^{\star}\left(\mathbf{G}_{\ell+1}\right)\right),\mathbf{Z}_{\ell}\right\rangle_{\mathcal{H}_{\ell}} \tag{21a}$$

$$= \left\langle\mathcal{T}_{\ell+1}^{\star}\left(\mathbf{G}_{\ell+1}\right),\left[\mathrm{D}\phi_{\ell}\left(\mathbf{Z}_{\ell}\right)\right]\left(\mathbf{Z}_{\ell}\right)\right\rangle_{\mathcal{H}_{\ell}} \tag{21b}$$

If $\phi_\ell$ is a positive-homogeneous of order $k_\ell$, then using Euler's theorem, we have

$$\langle \mathbf{G}_\ell, \mathbf{Z}_\ell \rangle_{\mathcal{H}_\ell} = k_\ell \cdot \langle \mathcal{T}_{\ell+1}^\star \left( \mathbf{G}_{\ell+1} \right), \phi_\ell \left( \mathbf{Z}_\ell \right) \rangle_{\mathcal{H}_\ell} \tag{22a}$$

$$= k_\ell \cdot \langle \mathbf{G}_{\ell+1}, \mathbf{Z}_{\ell+1} \rangle_{\mathcal{H}_{\ell+1}} \tag{22b}$$

Using this recursion, we have

$$\frac{\mathrm{d}}{\mathrm{d}t} \left\| \mathcal{T}_\ell \right\|_{\mathrm{HS}(\mathcal{H}_{\ell-1}, \mathcal{H}_\ell)}^2 = -2 \left( \prod_{l=\ell}^{L} k_l \right) \cdot \langle \mathbf{G}_L, \mathbf{Z}_L \rangle_{\mathcal{H}_L} \tag{23a}$$

$$= -2 \left( \prod_{l=\ell}^{L} k_l \right) \cdot \langle \nabla_{\mathbf{f}} \mathcal{L} \left( \mathbf{f}_{\boldsymbol{\theta}} \left( \mathbf{X} \right); \mathbf{Y} \right), \mathbf{f}_{\boldsymbol{\theta}} \left( \mathbf{X} \right) \rangle_{\mathcal{H}_L} \tag{23b}$$

with $k_L \coloneqq 1$.

**Corollary 1.** *Under the setup in [Theorem 2](),*

$$\frac{d}{dt} \mathrm{Tr} \left( \mathcal{B} \left( \mathcal{T}_\ell \left( t \right), \mathcal{T}_{\ell+1} \left( t \right) \right) \right) = -2 \left( k_\ell - 1 \right) \left( \prod_{l=\ell+1}^{L} k_l \right) \cdot \langle \nabla_{\mathbf{f}} \mathcal{L} \left( \mathbf{f}_{\boldsymbol{\theta}} \left( \mathbf{X} \right); \mathbf{Y} \right), \mathbf{f}_{\boldsymbol{\theta}} \left( \mathbf{X} \right) \rangle_{\mathcal{H}_L}$$
$$- 2\delta \cdot \mathrm{Tr} \left( \mathcal{B} \left( \mathcal{T}_\ell \left( t \right), \mathcal{T}_{\ell+1} \left( t \right) \right) \right) \tag{6}$$

**Proof.** We begin by noting that

$$\mathrm{Tr} \left( \mathcal{B} \left( \mathcal{T}_\ell \left( t \right), \mathcal{T}_{\ell+1} \left( t \right) \right) \right) = \left\| \mathcal{T}_\ell \right\|_{\mathrm{HS}(\mathcal{H}_{\ell-1}, \mathcal{H}_\ell)}^2 - \left\| \mathcal{T}_{\ell+1} \right\|_{\mathrm{HS}(\mathcal{H}_\ell, \mathcal{H}_{\ell+1})}^2 \tag{24}$$

The result follows from [Theorem 2]().

**Lemma 1.** *A positive-homogeneous function, $\phi_k \left( \cdot; c_-, c_0, c_+ \right) : \mathbb{R} \to \mathbb{R}$, of order $k$ takes the form*

$$\phi_k \left( \cdot; c_-, c_0, c_+ \right) = \begin{cases} c_- \left( -x \right)^k, & x < 0 \\ c_0, & x = 0 \\ c_+ x^k, & x > 0 \end{cases} \tag{8}$$

*where $c_\pm = \phi \left( \pm 1 \right) \in \mathbb{R}$, and $c_0 = \phi \left( 0 \right) \in \mathbb{R}$. Moreover, if $k \neq 0$, then $c_0 = 0$.*

**Proof.** Say $\phi : \mathbb{R} \to \mathbb{R}$ is a positive-homogeneous function of order $k$. Then, for any $x \in \mathbb{R}_{++}$,

$$\phi \left( x \right) = \phi \left( x \cdot 1 \right) = x^k \phi \left( 1 \right) \tag{25}$$

Similarly, for any $x \in \mathbb{R}_{--}$,

$$\phi \left( x \right) = \phi \left( -x \cdot -1 \right) = \left( -x \right)^k \phi \left( -1 \right) \tag{26}$$

For any $c > 0$,

$$\phi \left( 0 \right) = \phi \left( c \cdot 0 \right) = c^k \phi \left( 0 \right) \tag{27}$$

Therefore, $c_0 = 0$ if $k \neq 0$, and it is a free hyperparameter if $k = 0$.

**Lemma 2.** *A sequence of positively homogeneous operators can be replaced with one whose homogeneity order is the product of the homogeneity orders of the component operators.*

**Proof.** Say $\phi_1 : \mathcal{H}_0 \to \mathcal{H}_1$ is a positive-homogeneous function of order $k_1$ and $\phi_2 : \mathcal{H}_1 \to \mathcal{H}_2$ is one of order $k_2$. Then, for any $x \in \mathcal{H}_0$ and any $c \in \mathbb{R}_{++}$,

$$\phi_2 \left( \phi_1 \left( cx \right) \right) = \phi_2 \left( c^{k_1} \phi_1 \left( x \right) \right) = c^{k_1 k_2} \phi_2 \left( \phi_1 \left( x \right) \right) \tag{28}$$

That is, $\phi_2 \circ \phi_1$ is positive-homogeneous of order $k_1 k_2$.

## C  EXPERIMENTAL DETAILS

**Mean Squared-Error.** The MSE loss is computed as

$$\mathcal{L} \left( \mathbf{f}_{\boldsymbol{\theta}}; \mathbf{X}, \mathbf{Y} \right) = \frac{1}{2n} \left\| \mathbf{f}_{\boldsymbol{\theta}} \left( \mathbf{X} \right) - \mathbf{Y} \right\|_F^2 \tag{29}$$

Under this objective, the parameter-norm dynamics are computed using

$$-\nabla_{\mathbf{f}}\mathcal{L}\left(\mathbf{f}_{\boldsymbol{\theta}}\left(\mathbf{X}\right);\mathbf{Y}\right) = \frac{1}{n}\left(\mathbf{Y} - \mathbf{f}_{\boldsymbol{\theta}}\left(\mathbf{X}\right)\right) \tag{30}$$

**Cross-Entropy Loss.** Under the CE loss, the parameter-norm dynamics are computed using

$$-\nabla_{\mathbf{f}}\mathcal{L}\left(\mathbf{f}_{\boldsymbol{\theta}}\left(\mathbf{X}\right);\mathbf{Y}\right) = \frac{1}{n}\left(\mathbf{Y} - \mathsf{softmax}\left(\mathbf{f}_{\boldsymbol{\theta}}\left(\mathbf{X}\right)\right)\right) \tag{31}$$

where softmax is applied column-wise.

**Difference Quotient.** With learning rate $\eta$, the forward DQ is computed as

$$\mathrm{DQ}\left(t\right) = \frac{\left\|\boldsymbol{\theta}_{t+1}\right\|^2 - \left\|\boldsymbol{\theta}_t\right\|^2}{\eta} \tag{32}$$

**Figure Details:**

- Figure 1a. 3-layer CNN (last layer being a fully-connected readout) with 8 channels in the hidden layers, ReLU activation and Max pooling after each hidden layer, trained to minimize the MSE loss on CIFAR-10 (Krizhevsky, 2009) using gradient descent with learning rate $\eta = 0.04$.

- Figure 1b. 4-layer CNN with 4 channels in the hidden layers, LeakyReLU activation ($c = 0.1$), and BatchNorm – without affine transform – applied *only before the readout*, trained to minimize the CE loss on MNIST (LeCun et al., 2010) using $\eta = 0.02$.

- Figure 1c. 4-layer MLP with hidden layers of size 256, LeakyReLU activation, and LayerNorm (applied only before the readout), trained to minimize the CE loss on CIFAR-100 (Krizhevsky, 2009) using $\eta = 0.005$.

- Figure 1d. 4-layer MLP with hidden layer sizes 1024, 512 and 256, and Tanh activation, trained to minimize the MSE loss on CIFAR-100 using $\eta = 0.01$ and weight decay $\delta = 0.075$.

- Figure 2a and Figure 2b. Same as above, except with CE loss.

## D  ADDITIONAL FIGURES FOR TANH NETWORKS

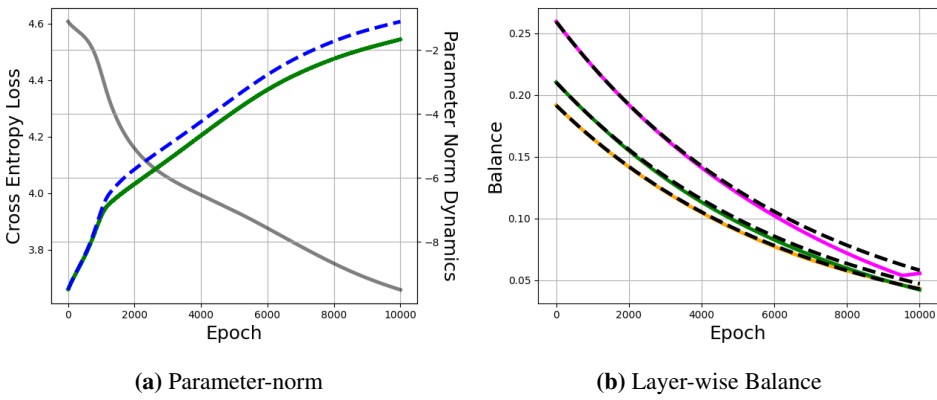

(a) Parameter-norm

(b) Layer-wise Balance

**Figure 2:** Theoretical predictions and empirical dynamics of for Tanh-MLPs trained with CE loss.

## E  GEOMETRIC INTERPRETATION

In this section, we look at the inner-product in Theorem 2 more closely, taking MSE loss as the running example for simplicity.

First off, we note that networks are usually initialized such that $2L/n \cdot \langle \mathbf{f}\left(\mathbf{X}\right), \mathbf{Y} - \mathbf{f}\left(\mathbf{X}\right)\rangle_F < 0$, so that parameter norm decreases early in training. In Figure 3, we plot the mean and standard deviation of this term, using the FashionMNIST dataset (Xiao et al., 2017), and Tanh-MLPs of

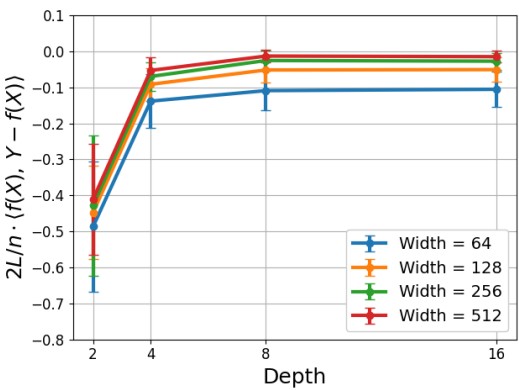

**Figure 3:** Parameter norm tends to decrease at initialization; Tanh-MLP with FashionMNIST dataset.

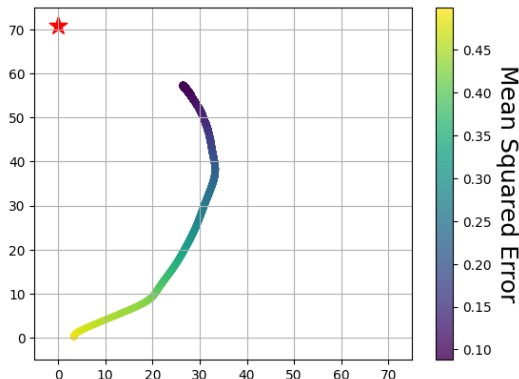

**Figure 4:** Projected trajectory of a LeakyReLU-MLP trained on the FashionMNIST dataset.

varying widths and depths initialized using Kaiming uniform initialization (He et al., 2015). It is clear to see that networks are initialized to decrease parameter norm, on average, but this effect reduces with increasing width, as well as depth.

Next, we visualize the dynamics of the network output by projecting it down to 2D. Specifically, we train a 4-layer LeakyReLU-MLP with hidden layers of size 256, trained to minimze MSE loss on FashionMNIST using $\eta = 0.02$. We project the labels $\mathbf{Y}$ down to $(0, \|\mathbf{Y}\|)$, as well as the outputs at each step such that their norm $\|\mathbf{f}(\mathbf{X})\|$ and the inner-products $\langle \mathbf{f}(\mathbf{X}), \mathbf{Y} - \mathbf{f}(\mathbf{X}) \rangle$ are preserved:

$$\mathbf{f}(\mathbf{X}) \mapsto \left( \sqrt{\|\mathbf{f}(\mathbf{X})\|^2 - \frac{\langle \mathbf{f}(\mathbf{X}), \mathbf{Y} - \mathbf{f}(\mathbf{X}) \rangle^2}{\|\mathbf{Y}\|^2}}, \frac{\langle \mathbf{f}(\mathbf{X}), \mathbf{Y} - \mathbf{f}(\mathbf{X}) \rangle}{\|\mathbf{Y}\|} \right) \tag{33}$$

We visualize the trajectory under this projection in Figure 4, where we can see the outputs first move radially out, and then slowly turn towards the labels.

