# OpenReview forum: "Unified Perspectives on Balancedness and Parameter-norm Evolution in Neural Nets"
_ICLR.cc/2026/Workshop/Sci4DL — Sci4DL 2026_

### Official Review · Reviewer_8QAE · 2026-02-18

**Fit:** 3
**Significance:** 2
**Confidence:** 2

**Summary:**

This paper proves the dynamics of layer-balancedness and parameter-norm in a unified and general framework. Key insights include demonstrating that consecutive layers without intervening nonlinearities converge exponentially toward a balanced state under weight decay and that normalization layers result in the sizes of all previous layers to decay at an exponential rate proportional to weight decay.

**Strengths:**

The framework of this paper is quite general and covers a wide range of modern components including pooling and normalization layers. It successfully recovers existing results from literature as special cases.

**Suggestions:**

While the paper provides a successful framework to unify existing results, the resulting insights—such as exponential convergence to balancedness—largely mirror existing literature. The paper's impact would be significantly strengthened if the authors could leverage this framework to uncover entirely novel insights or phenomena that were previously invisible to previous architecture-specific analyses.

---

### Official Review · Reviewer_GW9a · 2026-02-26

**Fit:** 2
**Significance:** 2
**Confidence:** 2

**Summary:**

This paper studies how the weights of a neural network evolve during training, focusing on two related phenomena: how consecutive layers become balanced, and how the norms of the layers grow or shrink over time. The authors  show that when two linear layers appear back to back, any imbalance between them shrinks exponentially under weight decay. They also derive a formula for how the squared norm of each layer changes during training, showing that all layers are driven by the same quantity involving the network’s output and the loss gradient. In other words, the way the weights grow or shrink is directly tied to how the model’s predictions change over time. This extends earlier results from linear networks to a much broader class of models.

**Strengths:**

The paper provides a clear and general theoretical framework that brings together several existing results on layer balancing and norm growth, rather than treating each architecture separately, making the paper more cohesive

The framework is broad enough to include fully connected layers, convolutions, pooling, etc. which extends prior analyses that were mostly limited to linear networks, and theoretical claims are supported with experiments that compare predicted norm dynamics to empirical trajectories, and the agreement is strong where gradient flow is expected to approximate gradient descent

Assumptions are carefully noted, and the paper clearly discusses where the theory does and does not apply, such as the breakdown near the edge of stability, which makes the claims feel more credible

**Suggestions:**

The theory is developed under gradient flow, but the experiments use gradient descent. Although the paper discusses the edge of stability, it would help to more clearly show when and how the discrete training dynamics start to deviate from the continuous predictions, for example by varying the learning rate and measuring when the mismatch appears

The main results rely on positive homogeneity assumptions for the nonlinearities. The experiments with tanh suggest the predictions sometimes still hold even when the assumptions are violated, but the paper does not fully explain why (though this makes sense as long as the activations are very close to 0, as tanh acts almost positively homogenous and consistent when the post-synaptic, pre-activation value is close to 0). However, a clearer discussion of when the theory is expected to apply, and when it is not, would strengthen the work

---

### Official Review · Reviewer_hKw7 · 2026-02-27

**Fit:** 2
**Significance:** 2
**Confidence:** 2

**Summary:**

This paper provides a more general framework (than prior works) for studying the dynamics of the balance between consecutive layers, and layer-wise parameter-norms. There are various results in the literature that tie norms of layers and balancedness (suitably defined) under gradient flow - this works extends and generalizes this analysis through an operator-level formulation and gets results beyond just linear MLPs, which is nice. Further, the paper finds one scalar (layer output inner product with loss gradient of it) that governs norm dynamics. Some simple experiments validate the theory.

**Strengths:**

The paper gives a mathematically clean framework, with some novel results that generalize to more architectures or layer types than prior work. THe general operator abstraction unifies several prior strands in the literature. Thus this paper gives a useful theoretical conceptualization of normalization of layers.

**Suggestions:**

One significant weakness is the homogeneity assumption, which doesn't necessarily hold in practice. Theorem 1 requires no non-linearity between layers, which is restrictive. It would be great to broaden this result. Lastly, as the authors state, gradient flow analysis tracks GD only up to the edge of stability. It would be nice to quantify until which point in time, say, the results still hold.

---

### Meta-Review · Area_Chair_qZT8 · 2026-02-28

**Recommendation:** Accept

**Metareview:**

This paper generalizes previous studies of balancedness between different layers during gradient flow training in deep non-linear networks. Reviewers praised the theoretical results, and authors relied on experiments to verify them beyond the regime of their assumptions. I recommend acceptance.

---

### Decision · Program_Chairs · 2026-03-02

Accept